# Time Series Data Provide Insights into the Evolution and Abundance of One of the Most Abundant Viruses in the Marine Virosphere: The Uncultured Pelagiphages vSAG 37-F6

**DOI:** 10.3390/v16111669

**Published:** 2024-10-24

**Authors:** Marina Vila-Nistal, Ramiro Logares, Josep M. Gasol, Manuel Martinez-Garcia

**Affiliations:** 1Department of Physiology, Genetics, and Microbiology, University of Alicante, Carretera San Vicente del Raspeig, San Vicente del Raspeig, 03690 Alicante, Spain; marina.vila@ua.es; 2Multidisciplinary Institute for Environmental Studies (IMEM), University of Alicante, Carretera San Vicente del Raspeig, San Vicente del Raspeig, 03690 Alicante, Spain; 3Institut de Ciències del Mar (ICM-CSIC), 08003 Barcelona, Spain; ramiro.logares@icm.csic.es (R.L.); pepgasol@icm.csic.es (J.M.G.)

**Keywords:** viruses, *Pelagibacter*, SAR11, single-virus genomics, vSAG 37-F6, pelagiphage, metagenomics

## Abstract

Viruses play a pivotal role in ecosystems by influencing biochemical cycles and impacting the structure and evolution of their host cells. The widespread pelagiphages infect *Pelagibacter* spp., the most abundant marine microbe on Earth, and thus play a significant role in carbon transformation through the viral shunt. Among these viruses, the uncultured lytic pelagiphage vSAG 37-F6, uncovered by single-virus genomics, is likely the most numerous virus in the ocean. While previous research has delved into the diversity and spatial distribution of vSAG 37-F6, there is still a gap in understanding its temporal dynamics, hindering our insight into its ecological impact. We explored the temporal dynamics of vSAG 37-F6, assessing periodic fluctuations in abundance and evolutionary patterns using long- and short-term data series. In the long-term series (7 years), metagenomics showed negative selection acting on all viral genes, with a highly conserved overall diversity over time composed of a pool of yearly emergent, highly similar novel strains that exhibited a seasonal abundance pattern with two peaks during winter and fall and a decrease in months with higher UV radiation. Most non-synonymous polymorphisms occurred in structural viral proteins located in regions with low conformational restrictions, suggesting that many of the viral genes of this population are highly purified over its evolution. At the fine-scale resolution (24 h time series), combining digital PCR and metagenomics, we identified two peaks of cellular infection for the targeted vSAG 37-F6 viral strain (up to approximately 10^3^ copies/ng of prokaryotic DNA), one before sunrise and the second shortly after midday. Considering the high number of co-occurring strains of this microdiverse virus, the abundance values at the species or genus level could be orders of magnitudes higher. These findings represent a significant advancement in understanding the dynamics of the potentially most abundant oceanic virus, providing valuable insights into ecologically relevant marine viruses.

## 1. Introduction

Viruses play a major role in the structure and evolution of ecosystems by both influencing the biogeochemical cycles [1] and affecting the abundance and distribution of their host cells [2]. In marine ecosystems, viruses are responsible for the lysis of 10–20% of the bacterial community on a daily basis [3]. This high lysis rate transforms organic particulate matter into dissolved organic matter (i.e., the viral shunt) in a magnitude estimated to approach 10 billion tons of carbon per day; therefore, the associated release of inorganic nutrients represents a fundamental step in nutrient cycling and fuels the productivity of the oceans [4,5,6,7].

Various studies highlight the significant role of viruses in shaping the populations of key prokaryotic and eukaryotic species in the marine ecosystem. Over the last two decades, a wealth of studies on marine virology have documented the amazing global diversity of DNA viruses from pole to pole, including their host range, and infection dynamics on the most relevant primary and secondary producers of the oceans, such as the cyanobacteria *Prochloroccous* [8], *Emiliania huxleyi* [9], diatoms [10], or SAR11 [11]. Moreover, we are only at the initial stages of unraveling the complexities of the marine RNA virosphere [12].

Among the different marine viruses discovered using various omics approaches, the uncultured lytic pelagiphage vSAG 37-F6 uncovered by single-virus genomics appears to play a crucial role in the marine ecosystem [13,14]. This virus was estimated to be the most prevalent in the *Tara* ocean viromic dataset [15] and one of the most microdiverse viruses on Earth [16], with one of its capsid proteins being the most abundant viral protein in open oceans [14]. The viral strain vSAG 37-F6 was observed to infect ≈10–400 *Pelagibacter* cells per milliliter of seawater [17,18]. *Pelagibacter* spp. is the most prevalent microorganism on the planet, representing 30–40% of total surface ocean bacterial cells and making a substantial contribution to total marine biomass and the decomposition of marine dissolved organic matter [19,20]. Previously, it has been estimated in a sample from the Mediterranean Sea that the total number of infected cells by vSAG 37-F6 per mL ranged from 10 to ≈400, which meant a total potential C release from 124 fg to 4.9 pg [18] (assuming total C cell content in oceanic bacterial assemblages of 12.4 fg as described in the study by Fukuda and collaborators) [21]. Considering the ubiquity and high abundance of the virus vSAG 37-F6 and other co-occurring closed related viruses belonging to the same vSAG 37-F6 genus (hereinafter referred to as “vSAG 37-F6-like viruses”, sharing ≈70–95% of nucleotide genome identity) in all *Tara* samples collected from the tropical and subtropical ocean [14], and even its recent detection in the coldest seawater in Earth in Antarctic waters [22], it suggests that this virus has the potential to transform an enormous amount of carbon through the viral shunt, making it a major contributor to the marine carbon cycle [17,23].

Our understanding of the complex interactions between marine viruses and hosts is still in its early stages. While some studies inspected the time series analysis of the diversity and dynamics of ecologically important viruses in various marine sites, providing insights into infection dynamics and local and global effects on the marine ecosystem [4,24,25,26,27], little is known about vSAG 37-F6 despite its significant role in the abundance, diversity, and temporal dynamics of SAR11 [23,28]. For instance, a recent gene flow analysis has revealed the impact of genetic recombination in shaping vSAG 37-F6-like viruses and *Pelagibacter* host speciation [16]. To this date, and even though the diversity and spatial distribution of vSAG 37-F6 have been explored in several sites [16], only one study has investigated the temporal dynamic of this virus, but it was only over a 24 h period and using an indirect methodology consisting of calculating viral abundance through transcriptomic read mapping [17]. This lack of knowledge hinders our comprehension of this viral ecological model and the subsequent predictions concerning viral and host responses to global events such as climate change, denoting the need for further research on this specific topic. The objective of this study was to address this knowledge gap by exploring the temporal dynamics of vSAG 37-F6 and its temporal variability in order to determine the periodic fluctuations in abundance on a daily and seasonal basis and the long-term evolutionary patterns of this virus over a span of 7 years, providing valuable biological insights into the marine vSAG 37-F6 viral model.

## 2. Materials and Methods

### 2.1. Sample Collection and Processing

In this study, three different sets of samples from the epipelagic zone were used to perform temporal studies at different time scales: a monthly 7-year time series, an annual monthly study, and a fine-resolution (every 2 h) 24 h sampling (Figure 1).

For the 24 h and the annual study, the samples were collected from Cape Huertas in the Northwestern Mediterranean Sea (38.35345133629738, −0.403225894080336, at surface level).

For the annual study, samples from 10 to 20 L were collected in 1-month intervals from September 2020 to September 2021. All samples were filtered through a 0.2 µm membrane filter, and then the viruses present in the filtrate were concentrated to 20 mL using tangential flow filtration with a 30 kDa ployethersulfone Vivaflow 200 membrane (Sartorius, Göttingen, Germany). The concentrated samples were filtered again through a 0.2 µm membrane to remove any cell remaining in the samples, and the viral fractions were further concentrated to 250 µL using Amicon Ultra 0.5 mL centrifugal filter units (Merck Millipore, Burlington, MA, USA). To ensure the elimination of all external nucleic acids, each 200 µL concentrated sample was treated with 2 U of Turbo DNase I (ref. AM2238, Thermo Fisher Scientific, Waltham, MA, USA) at 37 °C for 1 h, followed by inactivation with 22.52 µL of inactivation buffer at room temperature for 5 min. Then, the viral fraction was recovered by centrifuging the samples at 9200× *g* for 1.5 min. For the nucleic acid extraction, the supernatant was treated with 1% proteinase K (ref. EO0491, Thermo Fisher Scientific) and 10% TE 10X at 65 °C for 1 h with agitation and inactivated by a 5 min ice incubation, and then the DNA was extracted with a MinElute Virus Spin Kit (Qiagen, Hilden, Germany) according to the protocol of the manufacturer. The quantity of extracted DNA was determined using a Qubit fluorometer (Thermo Fisher Scientific).

For the 24 h study, the samples were collected in 2 h intervals on the 20th of December 2021. The sample consisted of 1.1 L of seawater, which was split into two duplicates that were filtered through a 0.2 µm nitrocellulose filter (ref. WHA7182002, Whatman, Maidstone, UK), and 100 µL was filtered through a 0.2 PES syringe filter (ref. SLFG85000, Merck) to remove the cells present in the sample and was fixed using glutaraldehyde to a final concentration of 4%. The filters and the fixed samples were kept in dry ice until arrival to the laboratory and then frozen at −80 °C. The next day, the filters were processed to extract the DNA for metagenomics using the DNeasy PowerSoil Pro Kit (ref. 47014, Qiagen), and the total amount was measured using a Qubit fluorometer. For the fixed samples, 1 mL of each sample was transferred to a 0.02 µm filter membrane (ref. WHA68096002, Sigma-Aldrich, St. Louis, MO, USA) and rinsed with 10 mL of sterile milliQ water to remove the excess of glutaraldehyde. Finally, the filters were stained by SYBR Gold (ref. S11494, Thermo Fisher Scientific), and the total number of viruses was counted by epifluorescence microscopy.

For the 7-year time series study, metagenomic samples were collected from the Blanes Bay Microbial Observatory (BBMO) in the Northwestern Mediterranean Sea (41°40′ N, 2°48′ E; ≈1 km offshore) from the 13th of January of 2009 to the 15th of December of 2015. Each sample consisted of 6 L of seawater that was pre-filtered through a 200 μm nylon mesh and then filtered through a 20 μm nylon mesh, a 3 μm pore size 47 mm diameter polycarbonate filter, and then through a 0.2 μm pore size Sterivex unit (Millipore, Burlington, MA, USA). DNA extractions from the 0.2–3 μm fractions were performed using a phenol–chloroform protocol [29] and then purified using Amicon units (Millipore). The extracted nucleic acids of each sample were quantified with a NanoDrop-1000 spectrophotometer (Thermo Scientific).

### 2.2. Digital PCR

To quantify the number of vSAG 37-F6 viral particles present in each sample from the 24 h and annual series, we performed a chip-based Taqman digital PCR (dPCR) using a probe and primer set for the vSAG 37-F6, previously designed and validated by Martinez-Hernandez et al. (2019) [30] (37-F6 ddSeq4; Fw: TGTGTACCTTCACCCACTTG, Rv: AGAACCATCAGGAACTCTGTTAC, Pb: TGACCAGCTTGAACCACAATACCCA), following the protocol described in McMullen et al. (2019) [18]. The reactions were loaded on separate QuantStudio™ 3D Digital PCR 20K Chip Kit v2 instruments, using the genome of vSAG 37-F6 obtained by MDA as a positive control and MQ water as a negative control [14], and they were run in a thermocycler GeneAmp PCR System 9700 following the PCR conditions described in McMullen et al. (2019) [18]. Finally, the chips were processed in a QuantStudio 3D Digital PCR System as described in McMullen et al. (2019) [18]. For the annual dataset, all samples presented a dPCR precision (i.e., understood as the size of the confidence interval necessary for differentiation between two sample concentrations at a specified level of confidence) under 10% except for the samples collected in June (13.67%) and October (26.19%). For the 24 h study, all samples presented a precision under 15% except for the samples taken at 5:00 am (16.43%), 7:00 am (22.41%), and 11:00 am (25.40%).

### 2.3. Sequencing and Read Treatment

Libraries from the annual metaviromes were created using the Illumina DNA Prep Kit (ref. 20018705, Illumina, San Diego, CA, USA) and were sequenced using a NextSeq sequencer 250PE and a NovaSeq6000 150PE in Macrogen (Seoul, Rep. de Corea) and CNAG-CRG (Barcelona, Spain), respectively. For the 7-year time series, the samples were sequenced in CNAG-CRG using a HiSeq 4000 System 150PE (samples from 2009 to 2011) and a NovaSeq6000 150PE platform (samples from 2012 to 2015).

The reads were quality-filtered using fastp (v 0.23.2) [31] with the following parameters: -l 50 --detect_adapter_for_pe -r --cut_right_window_size 4 --cut_right_mean_quality 30 -c. For the 1-year temporal series, each metavirome was individually assembled using SPAdes (v. 3.13.0) [32] with the options: --meta -k 21,33,55,77,99,127 --only-assembler. For the 7-year temporal series, each metagenome was individually assembled using MEGAHIT (v. 1.2.9) [33] using the default setting. All assembled contigs with less than 500 bp were removed from the analyses, and the prediction of ORFs was performed with Prodigal (v. 2.6.3) [34] using the option -p meta.

### 2.4. Bioinformatic Analysis

To study the changes in the vSAG 37-F6 viral population, we quantified the number of viruses present in each sample of the annual dataset, mapping the metaviromic reads and contigs against the reference genome of vSAG 37-F6 using BLASTn with two thresholds: 70% identity and 50% query coverage to target viruses that were 37-F6-like at family/genus-level; and 95% identity and 85% query coverage to target viruses that belonged to the same species as vSAG 37-F6. Only the hits above these thresholds were taken into consideration. The number of reads recruited by the vSAG 37-F6 genome in each sample was normalized by the viral genome length and the virome length using R software (v. 4.0.2) as follows: Kpb recruited/(Kb viral length * Gb virome size) (KPKG).

To compare the quantification using dPCR and KPKG, we conducted a read recruitment analysis of the annual time series against the same gene used for designing the dPCR primers. We applied a threshold of 100% identity and 100% query coverage to mimic the stringency of dPCR.

To further see changes in the composition of the population over longer periods of time, we performed SNPs and dN/dS analysis with the 7-year time series dataset obtained from the BBMO. First, we mapped the metagenomic reads to the genome of vSAG 37-F6 using bowtie2 (v 2.4.2) [35] with the --very-sensitive option, then we created and compared individual profiles for each sample using inStrain (v 1.7.5) [36]. The tridimensional models of predicted proteins from vSAG 37-F6 were downloaded from the AlphaFold Protein Structure Database and analyzed using Swiss-PdbViewer (v 4.1).

All statistical analyses were performed in R software (v 4.0.2).

## 3. Results and Discussion

In this study, we analyzed the population structure and dynamics of the virus vSAG 37-F6 at different time scales in the Mediterranean Sea: 24 h, 1 year, and 7 years. In each study, different approaches were used, including mainly metagenomics, viromics, and absolute quantification by digital PCR (Figure 1).

### 3.1. Population Structure of Virus vSAG 37-F6 in a 7-Year Time Series from the Mediterranean Sea

First, using a comprehensive 7-year monthly time series picoplankton dataset from the BBMO, we analyzed the distribution of SNPs in the vSAG 37-F6 genome recovered from each prokaryote metagenomic sample. For this, we first mapped the metagenomic reads against the vSAG 37-F6 genome, created a genomic consensus profile for each sample using inStrain software (v. 1.7.5) [36], and then searched for biologically meaningful patterns of SNPs over the time series experiments (Figure 2A). It is important to remark that in the BBMO dataset, we studied mainly the viruses present in the prokaryote fraction that were likely active and infecting cells. According to the parameters used in the inStrain program, here in this section, we are assessing the population viral structure of vSAG 37-F6 at the species level (i.e., viral strains sharing >95% of nucleotide identity).

We found that the pattern of SNPs were similar between different months, with nucleotide diversity, defined as the average number of nucleotide differences per site between two DNA sequences in all possible pairs in the sample population, ranging from 0 to 0.12. These low numbers support the findings of previous works about the coexisting diversity being maintained over a long period of time [16]. When examining SNPs per gene, all genes present a negative selection (dN/dS between 0 and 0.8), as has been the case for other environmental viruses, such as in the deep sea [37,38,39], which implies that throughout the genome, there is a necessity to preserve the integrity and functionality of essential genes, ensuring that mutations that compromise their function are purged. Most non-synonymous mutations occurred in hypothetical proteins, suggesting that they may be linked to the regulation of host metabolism to promote viral replication as they exhibit lower conservation compared with structural and genome replication proteins [40]. Further investigation into the distribution of SNPs per position in the vSAG 37-F6 genome revealed only three regions that were free from non-synonymous SNPs between the nucleotide positions 3528–3846, 5503–5878, and 6738–7093, with coverage exceeding 50X, which rules out sequencing errors or hyper-variable islands and suggests that they are highly conserved and likely crucial for virus function (see Appendix A). These regions primarily consist of structural proteins and a hypothetical gene. In order to examine the non-synonymous mutations occurring repeatedly in these highly conserved genes over the span of 7 years, we explored the location of SNPs with the highest number of non-synonymous mutations in the capsid protein and chaperonin (Figure 2B), two proteins that are highly conserved, and we observed that they predominantly occur in regions with low conformational restrictions, not essential for protein folding and function.

We also calculated the virus abundance for each month (Figure 2C) and observed a recurring pattern of two peaks per year: one in spring and the other in the summer–fall season. These abundance peaks likely result from a myriad of ecological factors that increase the host abundance, and likely in turn, the abundance of their uncultured viruses. However, because the number of *Pelagibacter* species and strains infected by vSAG 37-F6 virus are still unknown, discerning the specific factors responsible remains a challenge. The decreases in abundance correlated with seasons of higher solar irradiance and temperature, also accompanied by a reduced input of dissolved organic matter from freshwater streams, which supposes a greater stress condition [41]. This viral population behavior mirrors a similar dynamic in their putative host SAR11 [42], which has also been previously described in other environments [12,43].

Lastly, we examined the viral diversity across all samples to determine if the month in which the abundance of virus vSAG 37-F6 peaked was attributed to the reappearance of the same viral strain or group of strains. For this purpose, we took the unique pattern of SNPs for each month to define the meta-population strains that were present in that moment. When comparing the average nucleotide identity (ANI) between each pair of samples, we observed that the ANI value remained constant regardless of the time elapsed between these samples (Figure 3), indicating a steady level of nucleotide-level genomic similarity over time. While previous studies have extensively documented the significant diversity of these viruses [14,16], our research reveals the persistence of this diversity within a temporal evolutionary framework. Subsequently, we compared the number and distribution of shared SNPs among sample pairs and found no consistent pattern of a specific strain being present across multiple months. This suggests that the peaks in abundance are not a result of the resurgence of the same strain but are the result of the emergence of new strains that configure a different SNP pattern.

### 3.2. Annual Variation of Virus vSAG 37-F6

To further increase our knowledge of the temporal dynamics of vSAG 37-F6, we studied the abundance of free viral particles in seawater in a once-a-month sampling for a total period of one year. Then, we extracted the viral fraction and quantified the copies of vSAG 37-F6 in each month using two different methods: read mapping and KPKG calculation and absolute quantification with dPCR. The primer design for digital PCR processing was very specific, targeting the likely strain vSAG 37-F6 originally described by single-virus genomics. However, considering the high diversity of these viruses and the high co-occurrence of viral strains in the same sample described in a previous study [15], we cannot rule out that other closely related strains are detected. Therefore, in this dPCR experiment, we are targeting vSAG 37-F6 mainly at the strain level, although other strains belonging to same species could be considered as well.

When comparing the abundance patterns with those previously described in the seven-year time series dataset, we similarly observed with the in silico approach two abundance peaks in the seasons of winter and fall (Figure 4). In the same manner, decreases in abundance coincided with months characterized by higher solar irradiance and higher stress conditions, which might limit the primary and secondary production and thus the nutrients available for bacterial growth [44].

When comparing in detail the abundances obtained by the two different methods, the mapping of the reads to the vSAG 37-F6 genome, and the absolute quantification using dPCR, we observed some differences in the viral abundance for the same sample depending on the technique due to the different sensitivity and specificity of each one (Figure 4B,C). While read mapping was performed with an identity and query coverage values were designed to detect the reads from all vSAG 37-F6 genus-like (identity 70%, query coverage 50%) viruses, the primers used for dPCR are very specific and only putatively inform us at the species or strain level. Based on the high level of population diversity that the vSAG 37-F6 virus presents [14], we speculate that the differences in abundance between the two methods are due to small variations in the composition of the population and in its diversity. This reflects the disparity between the consensus genome from which the primers for the dPCR had been designed and the actual genomes of the microdiverse population that exist at the specific site and time of sampling.

To test this hypothesis, we conducted a read recruitment for the gene ORF 22 of the vSAG 37-F6 genome targeted by the dPCR primers, employing very stringent in silico thresholds (see methods) to assess if this quantification, closer to the genotype identified by dPCR, yielded distinct results compared with using the entire genome. Our findings reveal that there was no significant correlation between both methods. Although a weak correlation was observed between the KPKG measured via whole genome sequencing and the abundance of gene 22 (R^2^ = 0.51), suggesting that this gene contributes to the overall viral abundance, it is evident that many other genotypes not captured by dPCR also significantly contribute to total abundance (Appendix A). Hence, the combination of these two methodologies complements the advantages and disadvantages inherent to each of them, as the high specificity of dPCR in combination with a less stringent method such as read mapping can prove advantageous when studying populations exhibiting elevated rates of diversity.

### 3.3. Fine-Resolution Diel Cycling of Virus vSAG 37-F6

To assess changes in abundance over a 24 h cycle, we sampled seawater every 2 h, filtered it through a 0.2 μm filter, and processed it to extract the DNA and sequence the prokaryote metagenome of each sample and quantify the absolute number of virus vSAG 37-F6 copies in the prokaryotic pico-sized fraction through dPCR. Part of the sample was fixed with glutaraldehyde and stained to count the total number of free viruses.

The abundance of virus vSAG 37-F6 at the strain level was between 0.18 and 4.4 copies/mL of seawater and from 18.62 to 989.25 copies/ng of DNA of the prokaryotic fraction. The results of the diel cycling indicate a primary peak of 1.98 copies of vSAG 37-F6/mL of seawater and 989.25 copies/ng of the DNA of the prokaryotic fraction at midday (15:00), and the data suggest a secondary less distinct peak of ≈1 copy/mL of seawater and 93.7 copies/ng of the DNA of the prokaryotic fraction before dawn (3:00–5:00) (Figure 5). The number and time of the peaks agrees with results obtained in a similar study at Osaka Bay, which was based on using transcriptomic reads [17]. In this case, information about host abundance is not available; it is likely also similar to the one in the work based on the parallelism between viral dynamics, as viral abundance commonly closely follows the abundance of their primary hosts [42]. As mentioned before, the ecological factors responsible for this abundance peak are still unclear. We also would like to emphasize that the sampling was conducted in December, a month with low solar irradiance, so it could be possible that in warmer months, wherein this stress condition is more important, the second peak would be more delayed during the day.

## 4. Conclusions

This study increases our knowledge about the temporal dynamics of one of the putatively most numerous viruses in the temperate oceans, from the long-term temporal series to the study of its diel cycle. The analysis of the seven-year time series dataset revealed that all genes exhibited negative selection, with varying degrees of pressure among different genes. Most non-synonymous SNPs occurring in structural proteins were situated in regions with low conformational restrictions. Notably, three regions showed no non-synonymous mutations. We also found that the pattern of abundance of the vSAG 37-F6 virus observed in this 7-year dataset displayed two peaks during early spring and fall, while declines coincided with months characterized by elevated UV radiation and overall stress conditions, closely mirroring the abundance pattern of the putative host SAR11. Remarkably, upon comparing the number and distribution of shared SNPs across all months, we discovered that the peaks of abundance corresponded to the emergence of new viral strains. Furthermore, when analyzing the SNP patterns, we found that these new strains were highly similar to those that already existed, as all strains presented a high ANI regardless of the time elapsed between any two samples. When studying the annual dataset, we found a similar pattern of abundance to the one described in the 7-year dataset despite the difference in sampling location. The abundance annual dataset obtained by both metagenomics and dPCR revealed the importance of combining in silico and experimental approaches in marine virology to quantify and monitor a highly diverse population such as the vSAG 37-F6 virus to overcome limitations and biases. Lastly, when studying the diel cycling of the vSAG 37-F6 virus, we found two peaks of abundance, one less distinct at the hours prior to the sunrise (3:00–5:00 am) and one more pronounced shortly after midday (15:00). Given the high specificity of dPCR, the data obtained with this method corresponds to a specific strain of the 37-F6 virus that is actively infecting its corresponding strain of *Pelagibacter* spp. The characterization of these strains and their respective hosts would represent a significant advancement in the study of the putatively most predominant virus in the ocean, thereby enhancing our insight of the aquatic environment.

While this study provides valuable insights into the temporal dynamics and genetic diversity of the vSAG 37-F6 virus, it presents some limitations that should be taken into consideration. For instance, the fact that the seven-year dataset was collected from a single sampling location may limit the ability to extrapolate these findings to other marine environments, raising the interesting question of whether similar results would be observed in time series data from different geographical regions. For the 24 h study, repeating the sampling in different seasons could help to confirm to which grade the effects of temperature and solar irradiance affect viral abundance and if there are other factors that similarly affect viral viability. Addressing these limitations in future studies could provide a more comprehensive perspective and strengthen the validity of the current conclusions.

## Figures and Tables

**Figure 1 viruses-16-01669-f001:**
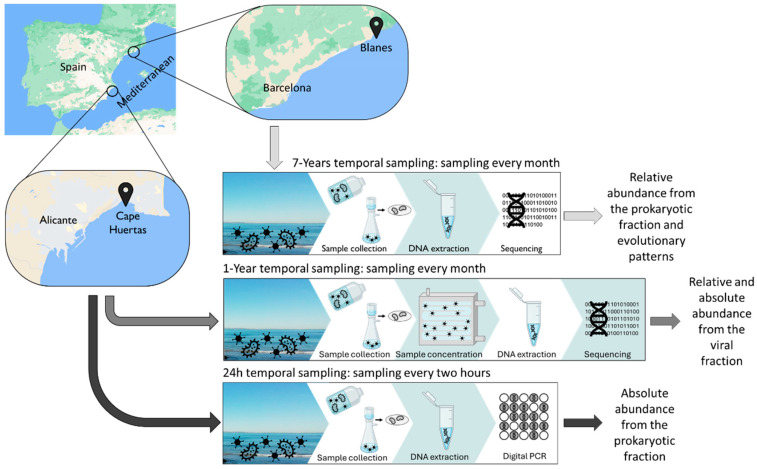
Geographical locations of the sampling points along with a summary of procedures for conducting the time series experiments and research questions addressed (text to the right in figure panels). Two different geographic locations were studied: seawater samples collected from Cape Huertas (Alicante, Spain) were used for the 1-year and 24 h temporal samplings, while seawater from the Blanes Bay Microbial Observatory (Blanes, Spain) was used for the 7-year temporal sampling. Different procedures were conducted as follows: for the case 7-year and 24 h temporal samplings, DNA from microbial fractions were obtained for sequencing (i.e., microbial metagenomes) and digital PCR, respectively. For the 1-year temporal sampling, the viral fraction was purified and the DNA extracted and sequenced, obtaining viral metagenomes.

**Figure 2 viruses-16-01669-f002:**
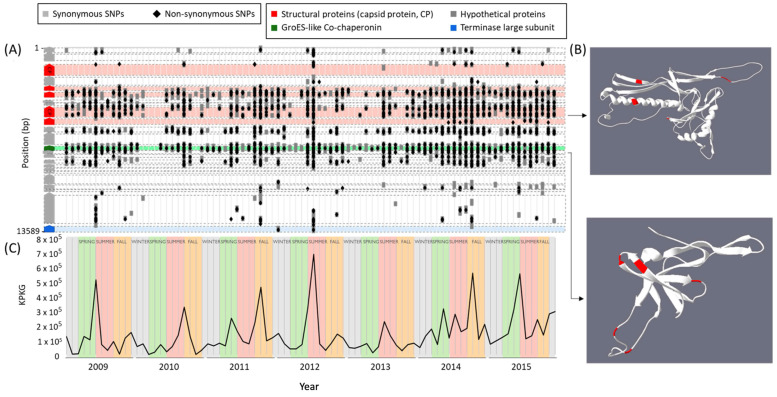
Temporal sampling across a 7-year time series. (**A**) Synonymous (grey) and non-synonymous (black) SNPs from the genomic consensus of each monthly sample and their respective positions within the 37-F6 genome. (**B**) Tridimensional structure of two proteins showing that the placement of the non-synonymous SNPs is primarily situated within the loops of the secondary structure. (**C**) Monthly measurement of viral abundance for 37-F6 over 7 years, categorized at the species level (black color), assessed using KPKG. CP: Capsid protein; SNPs: single-nucleotide polymorphisms.

**Figure 3 viruses-16-01669-f003:**
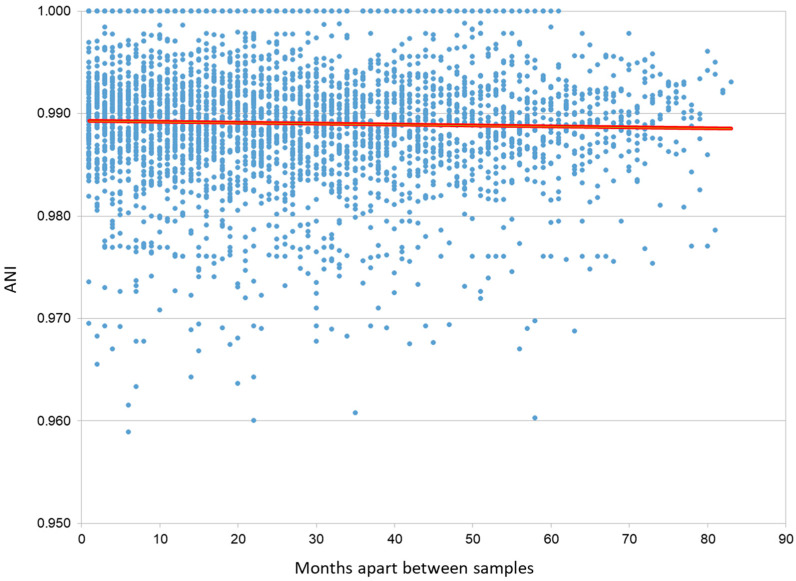
Comparison of the ANI among different sample pairs. Relationship between the ANI of two samples and their spatial difference. The red trend line is horizontal, indicating the independence of both parameters. This observation was confirmed by a Pearson’s correlation test, which yielded a coefficient of −0.02. ANI: Average nucleotide identity.

**Figure 4 viruses-16-01669-f004:**
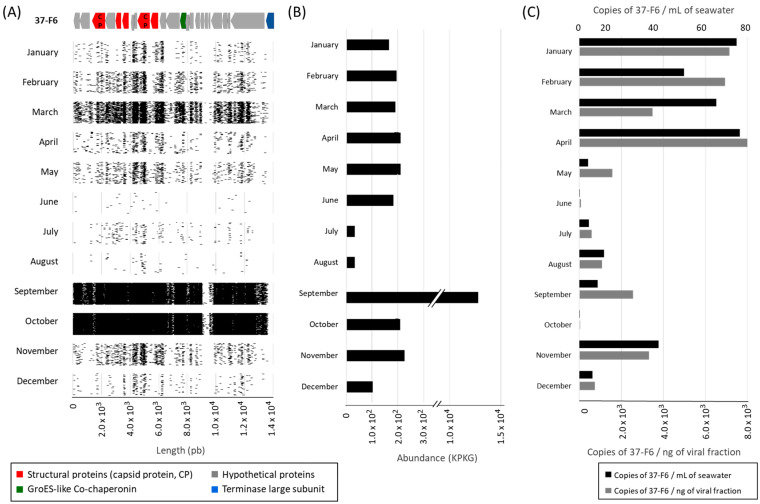
Annual pattern of the 37-F6 virus. (**A**) Mapping of the reads of each month against the 37-F6 genome; all reads are depicted as matched with an identity of over 70 and a query coverage of over 50 (at the viral genus level). (**B**) Abundance of the virus measured by the reads recruited by the 37-F6 genome normalized by the viral genome length and the virome length (KPKG). (**C**) Absolute quantification of 37-F6 by digital PCR, showing the number of viral copies per mL of seawater and ng of viral fraction. CP: Capsid protein; KPKG: kilobase of reads per kilobase of the genome per gigabase of reads.

**Figure 5 viruses-16-01669-f005:**
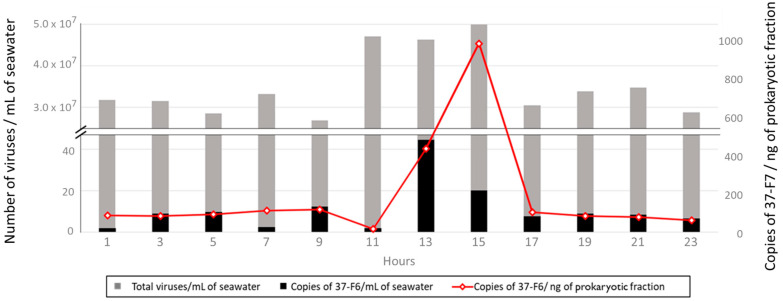
Diel cycling of the 37-F6 virus. Viral abundance represented and measured by PCR in number of copies of 37-F6 per mL of seawater (black) and copies of 37-F6 per ng of the prokaryotic fraction (red) compared with the total amount of viruses per mL of seawater (grey), measured by staining with SYBR Gold and counting using epifluorescence microscopy.

## Data Availability

The datasets generated and analyzed during the current study are available in the Zenodo repository under the following link: https://zenodo.org/records/10886232.

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
