# Peer review of "Time Series Data Provide Insights into the Evolution and Abundance of One of the Most Abundant Viruses in the Marine Virosphere: The Uncultured Pelagiphages vSAG 37-F6"

_viruses, 2024, doi:10.3390/v16111669_

Round 1

Reviewer 1 Report

Comments and Suggestions for Authors

The authors present an interesting and technically sound report on the temporal dynamics of the abundant vSAG 37-F6 pelagiphage. The  methods, data and conclusions presented are, to this reviewer, acceptable for publication.

Author Response

Answer (A): We are glad that this referee finds our paper acceptable for publication

Reviewer 2 Report

Comments and Suggestions for Authors

The uncultured phage vSAG 37-F6 lyses very abundant oligotrophic bacteria Pelagibacter spp., thereby contributing significantly to the carbon cycle. The study is aimed at closing the knowledge gap in understanding temporal dynamics of the uncultured pelagiphage vSAG 37-F6 and provide the insights into its ecological impact. The experimental data were obtained from three time series of seawater samples at the scales of hours, days and months.

According to the Abstract, the principal findings are related to (1) the discovery of negative selection in the long-term data series, (2) low genetic diversity within one-year samples, (3) seasonal abundance patterns, (3) the occurrence of amino acid changes in protein-coding regions with low conformational restrictions. The 24-hour sampling series allowed to (4) identify two peaks of cellular infections by the vSAG 37-F6 strain.

Major comments

The principal findings 1-4 are low-level research information, that should be evaluated in the context of existing knowledge and transformed into more complex conclusions. Is it usual for the genomes of marine viruses to undergo negative selection? Are there other scenarios? Can this finding be connected to the carbon cycle or something else? Assessed genetic diversity should be compared to estimates obtained by other authors. Consider if the insights into the connection between viral diversity and host ranges provided by Enav et al. [1] are relevant to your study. Explain what is expected and what is unexpected in the described temporal abundance patterns.

The paper claims “providing valuable insights into ecological marine viral models”. At this stage, it is completely unclear statement. Precisely what was found useful for modeling organic particulate matter dissolution by 37-F6-like viruses? According to the source [2], “The key processes to be represented in a microbial carbon pump model are those that describe the production and removal of recalcitrant dissolved organic carbon in different vertical regimes.” The source [3] constructed three models describing dissolved organic matter flux. The employed data concerned biomass, growth rates and phytoplankton death rate constants. Apparently, the current version of the MS does not contain related information. In the Introduction, set the relevance by mentioning quantitative models and geochemical questions that can benefit from your results. The Results and Discussion must have a paragraph with the data thought to be useful for ecological modeling.

The three time series of data were obtained with the use of different sampling and sample processing procedures. Currently, it’s almost impossible to understand precisely what was done in which case. Prepare a visualization explaining the study pipeline to highlight the heterogeneity of experimental data acquisition process depending on the time series type. Figure 4A is not informative and should be omitted.

Study limitations should be discussed in a separate paragraph.

In the paper, what relates to the uncultured pelagiphage vSAG 37-F6 and what relates to the 37-F6-like viruses? What is the difference between the two? The terms are used interchangeably, severely undermining scientific soundness of the MS.

How you can differentiate time factor from spatial factor, especially in the 24h study?

Minor comments

Evaluate your principal findings and decide if there is a possibility to prepare an informative title.

After reading the abstract, three immediate questions arise. What means the information about 10 thousand copies per nanogram of prokaryotic DNA, what it is related to? What high number of co-occurring strains we should consider, if in one season, all the viral strains are nearly identical? What is the difference between the population dynamics and temporal variability? Please amend the abstract so that these questions don’t appear.

The search for "microdiverse population" in Google Scholar retrieves 26 results. What lies behind the term? I find it redundant and distracting, especially in the Abstract.

Abstract, “up to 103 copies/ng”. The special symbol went lost. To avoid problems, you can omit it or substitute with the word “approximately”.

Check the provided sampling coordinates and their description. I put the provided Cape Huertas coordinates into an online converter [4] and obtained decimal coordinates 38.353972,0.426833. The point lies 72.40 kilometers (44.99 miles) away from Cape Huertas and 42.89 kilometers (26.65 miles) away from the nearest coast at Penyal d'Ifac. The same tool translates provided coordinates of the BBMO sampling site to 41.666667,2.800000. The point can be found approximately 780 meters offshore, in agreement with site description in the MS.

Mention the epipelagic zone as the source of samples, as it is an important term describing the relevance of your results [2].

Check for repeated uses of “this knowledge gap”, “most abundant”

The design of the figures should be made uniform. Take into account readability of the text and use standard spelling for the numbers.

Please use line numbers in further submissions

[1] Enav H, Kirzner S, Lindell D, Mandel-Gutfreund Y, Béjà O. Adaptation to sub-optimal hosts is a driver of viral diversification in the ocean. Nat Commun. 2018 Nov 8;9(1):4698. doi: 10.1038/s41467-018-07164-3. PMID: 30409965; PMCID: PMC6224464.

[2] Jiao N, Luo T, Chen Q, Zhao Z, Xiao X, Liu J, Jian Z, Xie S, Thomas H, Herndl GJ, Benner R, Gonsior M, Chen F, Cai WJ, Robinson C. The microbial carbon pump and climate change. Nat Rev Microbiol. 2024 Jul;22(7):408-419. doi: 10.1038/s41579-024-01018-0. Epub 2024 Mar 15. PMID: 38491185.

[3] Flynn KJ, Clark DR, Xue Y. MODELING THE RELEASE OF DISSOLVED ORGANIC MATTER BY PHYTOPLANKTON(1). J Phycol. 2008 Oct;44(5):1171-87. doi: 10.1111/j.1529-8817.2008.00562.x. Epub 2008 Sep 3. PMID: 27041714.

[4] https://coordinates-converter.com

Comments on the Quality of English Language

/

Author Response

REVIEWER 2

Major comments

The principal findings 1-4 are low-level research information, that should be evaluated in the context of existing knowledge and transformed into more complex conclusions. Is it usual for the genomes of marine viruses to undergo negative selection? Are there other scenarios? Can this finding be connected to the carbon cycle or something else? Assessed genetic diversity should be compared to estimates obtained by other authors. Consider if the insights into the connection between viral diversity and host ranges provided by Enav et al. [1] are relevant to your study. Explain what is expected and what is unexpected in the described temporal abundance patterns.

REPONSE: We appreciate this comment very much. The literature on negative or positive selection in marine viruses is still too limited to support any definitive statements, with only four papers to our knowledge addressing this issue, three of which also report negative selection in the viruses studied [1-3] and one reporting a positive selection [4]. As per the comparisons with studies from other authors, the work by Enav et al. focuses on generalist and sub-optimal viruses, making it too distinct from our study to provide meaningful insights for interpreting our results. Any further conclusions drawn from these other studies would be purely speculative, as we still lack a clear understanding of the host range of this virus and other critical factors that are being investigated in ongoing projects within our group.

  1. Castelán-Sánchez, H. G. et al. Extremophile deep-sea viral communities from hydrothermal vents: Structural and functional analysis. Marine Genomics 46, 16–28 (2019).
  2. Peng, Y. et al. Viruses in deep-sea cold seep sediments harbor diverse survival mechanisms and remain genetically conserved within species. ISME J 17, 1774–1784 (2023).
  3. Vlok, M., Lang, A. S. & Suttle, C. A. Marine RNA Virus Quasispecies Are Distributed throughout the Oceans. mSphere 4, e00157-19 (2019).
  4. Meng, L. et al. Genomic adaptation of giant viruses in polar oceans. Nat Commun 14, 6233 (2023).

The paper claims “providing valuable insights into ecological marine viral models”. At this stage, it is completely unclear statement. Precisely what was found useful for modeling organic particulate matter dissolution by 37-F6-like viruses? According to the source [2], “The key processes to be represented in a microbial carbon pump model are those that describe the production and removal of recalcitrant dissolved organic carbon in different vertical regimes.” The source [3] constructed three models describing dissolved organic matter flux. The employed data concerned biomass, growth rates and phytoplankton death rate constants. Apparently, the current version of the MS does not contain related information. In the Introduction, set the relevance by mentioning quantitative models and geochemical questions that can benefit from your results. The Results and Discussion must have a paragraph with the data thought to be useful for ecological modeling.

REPONSE: Corrected, the phrase about viral models was omitted. We thank this comment.

The three time series of data were obtained with the use of different sampling and sample processing procedures. Currently, it’s almost impossible to understand precisely what was done in which case. Prepare a visualization explaining the study pipeline to highlight the heterogeneity of experimental data acquisition process depending on the time series type. Figure 4A is not informative and should be omitted.

REPONSE: We have prepared a new figure to explain the protocols applied. Each time series addressed different objectives and required different procedures. I hope in the new version, this is clear. We thank this comment to improve our manuscript.

Study limitations should be discussed in a separate paragraph.

REPONSE: Added in the conclusions. We appreciate this comment because we think it strength our manuscript.

In the paper, what relates to the uncultured pelagiphage vSAG 37-F6 and what relates to the 37-F6-like viruses? What is the difference between the two? The terms are used interchangeably, severely undermining scientific soundness of the MS.

REPONSE: vSAG 37-F6 refers to the specific genome described in previous papers (Martinez-Hernandez et al., 2017 Nat Comm) while “37-F6-like” refers to the group of related viruses that can belong to different strain- and species-levels and therefore they show differences at the nucleotide identity level in comparison to the original described virus vSAG 37-F6. The use of “virus name-like” is widely extended in the literature to refer to similar viruses.

How can you differentiate time factor from spatial factor, especially in the 24h study?

REPONSE: As the sampling point was exactly the same for all experiments, all the differences between samples derive from the time factor

Minor comments

Evaluate your principal findings and decide if there is a possibility to prepare an informative title.

REPONSE: According to this referee, we have corrected the title to “Time series reveal negative selection on yearly emergent strains of one of the most abundant viruses in the marine virosphere: the uncultured pelagiphages vSAG 37-F6”

After reading the abstract, three immediate questions arise. What means the information about 10 thousand copies per nanogram of prokaryotic DNA, what it is related to? What high number of co-occurring strains we should consider, if in one season, all the viral strains are nearly identical? What is the difference between the population dynamics and temporal variability? Please amend the abstract so that these questions don’t appear.

REPONSE: In a previous publication of our group in ISME Journal (Martinez-Hernandez et al 2022), we characterize for the first time the number of co-occurring strains and species of vSAG37-F6 virus in the Mediterranean Sea. We reported the co-occurrence of up to ≈1,500 different viral strains (>95% nucleotide identity) and ≈30 related species (80-95% nucleotide identity) in a single oceanic sample. Unfortunately, our study is one of the first, but the first one in marine virology, to show empirical data about the number of co-occuring strains and species of an ecologically relevant and abundant uncultured virus in the ocean. Thus, the lack of data from other abundant viruses hinders our ability to discuss further on the question rised by this referee on “what is high or low”. We are not claiming here that all strains are nearly identical, and in fact there is variation as shown in Martinez-Hernandez et al 2022 ISME Journal.  The standard values In qPCR and digital PCR is number of genome copies or gene copy per ng of DNA. This is a common way to normalize the results that allow a cross comparison with further studies. We truly think it is very informative because it reports about the abundance of genome copies in seawater per ng of DNA extracted.

We have corrected all sentences and homogenize the text using only “temporal dynamics” for clarity.

The search for "microdiverse population" in Google Scholar retrieves 26 results. What lies behind the term? I find it redundant and distracting, especially in the Abstract.

REPONSE: Corrected, the apparition of the term in the abstract was substituted by “diversity”

Abstract, “up to 103 copies/ng”. The special symbol went lost. To avoid problems, you can omit it or substitute with the word “approximately”.

REPONSE: Corrected

Check the provided sampling coordinates and their description. I put the provided Cape Huertas coordinates into an online converter [4] and obtained decimal coordinates 38.353972,0.426833. The point lies 72.40 kilometers (44.99 miles) away from Cape Huertas and 42.89 kilometers (26.65 miles) away from the nearest coast at Penyal d'Ifac. The same tool translates provided coordinates of the BBMO sampling site to 41.666667,2.800000. The point can be found approximately 780 meters offshore, in agreement with site description in the MS.

REPONSE: Corrected, coordinates are 38.353972,-0.426833

Mention the epipelagic zone as the source of samples, as it is an important term describing the relevance of your results [2].

REPONSE: Added “In this study, three different sets of samples from the epipelagic zone were used…”

Check for repeated uses of “this knowledge gap”, “most abundant”

REPONSE: Corrected to avoid redundancies

The design of the figures should be made uniform. Take into account readability of the text and use standard spelling for the numbers.

REPONSE: Thanks for the suggestion

Please use line numbers in further submissions

REPONSE: Added

[1] Enav H, Kirzner S, Lindell D, Mandel-Gutfreund Y, Béjà O. Adaptation to sub-optimal hosts is a driver of viral diversification in the ocean. Nat Commun. 2018 Nov 8;9(1):4698. doi: 10.1038/s41467-018-07164-3. PMID: 30409965; PMCID: PMC6224464.

[2] Jiao N, Luo T, Chen Q, Zhao Z, Xiao X, Liu J, Jian Z, Xie S, Thomas H, Herndl GJ, Benner R, Gonsior M, Chen F, Cai WJ, Robinson C. The microbial carbon pump and climate change. Nat Rev Microbiol. 2024 Jul;22(7):408-419. doi: 10.1038/s41579-024-01018-0. Epub 2024 Mar 15. PMID: 38491185.

[3] Flynn KJ, Clark DR, Xue Y. MODELING THE RELEASE OF DISSOLVED ORGANIC MATTER BY PHYTOPLANKTON(1). J Phycol. 2008 Oct;44(5):1171-87. doi: 10.1111/j.1529-8817.2008.00562.x. Epub 2008 Sep 3. PMID: 27041714.

[4] https://coordinates-converter.com

Reviewer 3 Report

Comments and Suggestions for Authors

Thank you for your submission! This work dedicated to the analysis of the abundance patterns and microdiversity population of one of the most abundant viruses in the marine virosphere. This research work was conducted as part of the big science project being carry out by the authors of the article. Thank you for your very interesting work!

There is need for better understanding. Would you please clarify the following points.

1. “For the 24 hours and the annual study, the samples were collected from Cape Huertas in the north-western Mediterranean Sea (38o21’14.30’’ N, 0o25’36.60’’ E, at surface level)”. Please provide information about distance. How many kilometers offshore? It is important to exclude terrigenous origin of viruses.

2. “Martinez-Hernandez et al., (2019).” There are two papers in the reference list by Martinez-Hernandez et al., (2019): Martinez-Hernandez F, Fornas Ò, Lluesma Gomez M, Garcia-Heredia I, Maestre-Carballa L, López-Pérez M, et al. Single-cell genomics uncover Pelagibacter as the putative host of the extremely abundant uncultured 37-F6 viral population in the ocean. ISME J 2019; 13: 232–236 and Martinez-Hernandez F, Garcia-Heredia I, Lluesma Gomez M, Maestre-Carballa L, Martínez Martínez J, Martinez-Garcia M. Droplet Digital PCR for Estimating Absolute Abundances of Widespread Pelagibacter Viruses. Front Microbiol 2019; 10: 1226.

It is very interesting to know what did you used as positive and negative controls in your dPCR experiments? Please provide information for followers.

3.First, using a comprehensive 7-year monthly time series picoplankton dataset from the BBMO, we analyzed the distribution of SNPs in the vSAG 37-F6 genome recovered from each prokaryote metagenomic sample. It is very interesting to know about distribution Pelagibacter reads in each procaryote metagenomic sample. What is the relationship between host and virus? What do you know about coverage of Pelagibacter contigs and coverage of 37-F6 contigs in each metagenomic sample?

4.It is important to remark that in the BBMO dataset, we studied mainly the viruses present in the prokaryote fraction that were likely active and infecting cells.”

Is bacteriophage vSAG 37-F6 lytic or temperate? Temperate phage integrate their genome into the host chromosome and remain dormant.

5. Figure 1A. Please consider adding legend to the genome map.

6. Supplementary Figure 2A. Please explain. What is shown on the y-axis? COVERAGE of WHAT? All reads on 7-years sampling assembled in one contig? How coexist simultaneously low and high coverage in many position?

7. “Further investigation into the distribution of SNPs per position revealed only three regions free from non-synonymous SNPs, with coverage exceeding 50X, which rules out sequencing errors or hyper-variable islands and suggests that they are highly conserved and likely crucial for virus function (see supplementary Figure 2).” Please clarify position of this three regions.

8. “The absolute abundance of virus vSAG 37-F6 at the strain level was between 0.18 and 4.4 copies/mL …...” Absolute abundance? For 24h study samples was filtered through 0.2 µm filter. The next day, the filters were processed to extract DNA. It means that viral fraction < 0.2 µm were not included in the analysis?

9.”This study increases our knowledge about the temporal dynamics of one of the putative most abundant viruses in the temperate oceans, from the long-term temporal series to the study of its diel cycle.” But samples were collected from the Mediterranean Sea. Can we extrapolate the knowledge about the temporal dynamics?

Author Response

REVIEWER 3

“For the 24 hours and the annual study, the samples were collected from Cape Huertas in the north-western Mediterranean Sea (38o21’14.30’’ N, 0o25’36.60’’ E, at surface level)”. Please provide information about distance. How many kilometers offshore? It is important to exclude terrigenous origin of viruses.

REPONSE: The sample point was 0 km offshore. As Pelagibacter ubique and its pelagiphages are only of marine origin, there is no possibility of terrigenous viruses interfering in the analysis.

“Martinez-Hernandez et al., (2019).” There are two papers in the reference list by Martinez-Hernandez et al., (2019): Martinez-Hernandez F, Fornas Ò, Lluesma Gomez M, Garcia-Heredia I, Maestre-Carballa L, López-Pérez M, et al. Single-cell genomics uncover Pelagibacter as the putative host of the extremely abundant uncultured 37-F6 viral population in the ocean. ISME J 2019; 13: 232–236 and Martinez-Hernandez F, Garcia-Heredia I, Lluesma Gomez M, Maestre-Carballa L, Martínez Martínez J, Martinez-Garcia M. Droplet Digital PCR for Estimating Absolute Abundances of Widespread Pelagibacter Viruses. Front Microbiol 2019; 10: 1226.

REPONSE: Corrected to (2019b) and added to the bibliography

It is very interesting to know what did you used as positive and negative controls in your dPCR experiments? Please provide information for followers.

REPONSE: Added in the correspondent section ( “using the genome of vSAG 37-F6 obtained by MDA as a positive control and MQ water as negative control”).

“First, using a comprehensive 7-year monthly time series picoplankton dataset from the BBMO, we analyzed the distribution of SNPs in the vSAG 37-F6 genome recovered from each prokaryote metagenomic sample. It is very interesting to know about distribution Pelagibacter reads in each procaryote metagenomic sample. What is the relationship between host and virus? What do you know about coverage of Pelagibacter contigs and coverage of 37-F6 contigs in each metagenomic sample?

REPONSE: While mapping the reads against the genome of Pelagibacter is a possibility, the high diversity of this bacteria limits our ability to identify which species or strain is most abundant in this area, or whether that genome is actually the host of the vSAG 37-F6 virus. Our research group is currently trying to determine the exact host range of vSAG 37-F6, which will help answer these questions in the future. We appreciate this interesting comment!.

Is bacteriophage vSAG 37-F6 lytic or temperate? Temperate phage integrate their genome into the host chromosome and remain dormant.

REPONSE: The bacteriophage vSAG 37-F6 is lytic. The  information has been added at the abstract and introduction.

Figure 1A. Please consider adding legend to the genome map.

REPONSE: It has been modified in the new version

Supplementary Figure 2A. Please explain. What is shown on the y-axis? COVERAGE of WHAT? All reads on 7-years sampling assembled in one contig? How coexist simultaneously low and high coverage in many position

REPONSE: the Y axis is the coverage, understood as the number of sequencing reads that are uniquely mapped to a reference, in this case the vSAG 37-F6 genome as explained in the figure caption. All reads of 7 years didn’t assemble in one unique contig. Positions have only one coverage value, the possible optic overlapping effect is due to the width of the bars, necessary for a clear graphic depiction

“Further investigation into the distribution of SNPs per position revealed only three regions free from non-synonymous SNPs, with coverage exceeding 50X, which rules out sequencing errors or hyper-variable islands and suggests that they are highly conserved and likely crucial for virus function (see supplementary Figure 2).” Please clarify position of this three regions.

REPONSE: It has been added in this same sentence in the new version.

“The absolute abundance of virus vSAG 37-F6 at the strain level was between 0.18 and 4.4 copies/mL …...” Absolute abundance? For 24h study samples was filtered through 0.2 µm filter. The next day, the filters were processed to extract DNA. It means that viral fraction < 0.2 µmwere not included in the analysis?

REPONSE: Correct, the prokaryotic fraction < 0.2 µm was not included in the analysis. The phrase “absolute abundance” was replaced to “abundance” for better clarity..

“This study increases our knowledge about the temporal dynamics of one of the putative most abundant viruses in the temperate oceans, from the long-term temporal series to the study of its diel cycle.” But samples were collected from the Mediterranean Sea. Can we extrapolate the knowledge about the temporal dynamics?

REPONSE: While the results are obtained from a specific sampling point, it has been shown that vSAG 37-F6 has similar behavior in different geographical locations (Martinez‐Hernandez et al., 2020). However, this point has been addressed in the limitations paragraph, where we acknowledge that the data were collected from the Mediterranean Sea, and that it would be valuable to compare these findings with time-series from other regions. Similar diel cycling pattern pattern was also found in the North Pacific Subtropical Gyre and the coast of Japan (please see Martinez-Hernandez et al (2020); Environ Microb Report. Although it seems that the diel cycle of this virus is very similar in the three different studied regions, we prefer to not extract definitive conclusion and be cautious in our statements.

Round 2

Reviewer 2 Report

Comments and Suggestions for Authors

During the revision, the Authors shifted the focus of the narration from ecological models to the temporal dynamics and genetic diversity of the vSAG 37-F6 virus and the fact its genomes undergo negative selection. In the previous round of peer review, I suggested a number of major improvements to the manuscript. As the performed revision does not address the comments in full, I perform an attempt to clarify the earlier concerns.

Major comments

The principal findings are low-level research information, that should be evaluated in the context of existing knowledge and transformed into more complex conclusions. Since the literature on negative or positive selection in marine viruses is too limited to support any definitive statements (and you have the selection mentioned in the title), please do find a way to expand on your findings. If nobody bothers to go further stating “positive” or “negative” software output, is there a significance of the type of selection at all? If you see it, share your understanding by explaining research gaps and mention contradicting evidence (Meng et al.). If there is nothing really important behind the selection type, shift the focus of the title to temporal dynamics. Current discussion of this particular issue seems to be satisfactory.

Figure 1. The visualization explaining the study pipeline to highlight the heterogeneity of experimental data acquisition process depending on the time series type should be improved.

Line 96. “For ... the annual study, the samples were collected from Cape Huertas”. Compare to Figure 1, where the yearly timeline has the input from the two locations. Should there be three instead of two pipeline charts?

In my perception, the figure has three parts, with only two parts explained in the caption. How you would describe the text to the right? Is it the type of research data or addressed question, or whatever? Decipher the ellipsis or remove the text after “evolutionary patterns”. In the caption, consider removing “the most relevant”.

The figure should show the frequency of sampling, not just state overall duration.

In the paper, what relates to the uncultured pelagiphage vSAG 37-F6 and what relates to the 37-F6-like viruses? What is the difference between the two? Add an explanation to the main text and check the paper for the correct use of the names. Beyond its introductory part, the abstract mentions only one specific virus, see Lines 19, 20, 30 and 34. However, one and seven-year time series are apparently related to the group of viruses.

Lines 66-68: How the general ubiquity of 37-F6-like viruses can suggest that one particular virus under the consideration has the potential to transform an enormous amount of carbon?

Minor comments

The search for "microdiverse population" in Google Scholar retrieves 26 results. What lies behind the term? Please search your text with the word “microdiverse”, if you agree with the proposed correction.

Check the provided sampling coordinates and their description again, or explain why the point 38.353972,0.426833 seems to be far from the point provided on the Figure 1. See also the compressed image in the attachment.

Take into account readability of the text on the figures and use standard spelling for the numbers. Check if Figures 3, 4 and 5 were substituted with identical graphics by a mistake. It is especially hard for me to read the axis labels of Figures 4 and 5. Compare them to the labels on Figure 3.

Figure 5. “Number of virus”. Please check.

Author Response

During the revision, the Authors shifted the focus of the narration from ecological models to the temporal dynamics and genetic diversity of the vSAG 37-F6 virus and the fact its genomes undergo negative selection. In the previous round of peer review, I suggested a number of major improvements to the manuscript. As the performed revision does not address the comments in full, I perform an attempt to clarify the earlier concerns.

We appreciate the time and effort of this referee and we are convinced that all comments are very useful to have an improved version of the manuscript.

Major comments

The principal findings are low-level research information, that should be evaluated in the context of existing knowledge and transformed into more complex conclusions. Since the literature on negative or positive selection in marine viruses is too limited to support any definitive statements (and you have the selection mentioned in the title), please do find a way to expand on your findings. If nobody bothers to go further stating “positive” or “negative” software output, is there a significance of the type of selection at all? If you see it, share your understanding by explaining research gaps and mention contradicting evidence (Meng et al.). If there is nothing really important behind the selection type, shift the focus of the title to temporal dynamics. Current discussion of this particular issue seems to be satisfactory.

ANSWER: We agree with this referee that the literature on negative selection in marine viruses is very limited with unclear conclusions about the meaning of negative selection and its impact on viral ecology. That is why we have decided to change the title to a more generalist title avoiding referring to the negative selection.  Now the title reads: “Time series data provide insights into the temporal dynamics and evolution of one of the most abundant viruses in the marine virosphere: the uncultured pelagiphages vSAG 37-F6

 Figure 1. The visualization explaining the study pipeline to highlight the heterogeneity of experimental data acquisition process depending on the time series type should be improved. The figure should show the frequency of sampling, not just state overall duration. Line 96. “For ... the annual study, the samples were collected from Cape Huertas”. Compare to Figure 1, where the yearly timeline has the input from the two locations. Should there be three instead of two pipeline charts?

ANSWER: Thanks for this comment. You are totally right and we forgot to include the pipeline for the 7-year time series. Apologize and thanks again. We have also included the frequency of sampling and more data in the new version of the figure.

In my perception, the figure has three parts, with only two parts explained in the caption. How you would describe the text to the right? Is it the type of research data or addressed question, or whatever? Decipher the ellipsis or remove the text after “evolutionary patterns”. In the caption, consider removing “the most relevant”.

ANSWER: thanks for this comment. We have expanded the figure caption to explain the figure adding more details. We have removed in the figure caption, according to this referee, “the most relevant”,  and also in the figure, the text after “evolutionary patterns”. In the figure caption we specify that the text to the right refers to addressed questions.

In the paper, what relates to the uncultured pelagiphage vSAG 37-F6 and what relates to the 37-F6-like viruses? What is the difference between the two? Add an explanation to the main text and check the paper for the correct use of the names. Beyond its introductory part, the abstract mentions only one specific virus, see Lines 19, 20, 30 and 34. However, one and seven-year time series are apparently related to the group of viruses.

ANSWER: Thanks for this comment that help us to improve the manuscript.  This terminology of “virus-like” has been widely used in many different examples with viruses that present high diversity, such as crassphage, which is one of the most abundant phages in the gut, and even in our previous publication in Nature Comm (Martinez-Hernandez et al. 2017), in which we describe the differences. Here, we have clarified in the new version of the manuscript the terminology of “vSAG-37-F6-like” at the introduction section and the differences with just vSAG 37-F6. In addition, we make clear at the beginning of each result section if we are addressing vSAG 37-F6 at the genus, species or strain levels.  “vSAG-37-F6-like” refers to those viruses that share 70-95% of nucleotide identity with the original vSAG 37-F6 described in Martinez-Hernandez et al 2017 (Nat Com) and that therefore likely belong to the same genus-level (i.e., different viral species). This is in agreement with the new ICTV guidelines and in line with the consensus statement published in Nat Biotechnology by Roux and colleagues (2019) and our previous publications (Martinez-Hernandez et al 2017). Thus, when we just use in the text the terminology “vSAG 37-F6”,  it means that the detected viruses from our experiments belong to the same species/strain-level. Here, in all of our analyses, we address this virus at the species level. But in some sentences, like the introduction, to be exact, and according to previous publications, we use the terminology of vSAG 37-F6-like (ln 78 introduction).

Lines 66-68: How the general ubiquity of 37-F6-like viruses can suggest that one particular virus under the consideration has the potential to transform an enormous amount of carbon?

ANSWER: We have clarified this sentence to: “Previously, it has been estimated in a sample from the Mediterranean Sea that the total number of infected cells by vSAG 37-F6 per mL ranged from 10 to ≈400, which meant a total potential C release from 124 fg to 4.9 pg (assuming total C cell content in oceanic bacterial assemblages of 12.4 fg as described in the study by Fukuda et al. (1998). Considering the ubiquity and high abundance of virus vSAG 37-F6 in all Tara samples collected from the tropical and subtropical ocean [14], and even its recent detection in the coldest seawater in Earth in Antarctic waters [20], suggest that this virus has the potential to transform an enormous amount of carbon through the viral shunt, making it a major contributor to the marine carbon cycle [23].

According to reported papers it is highly likely that the role of this virus on C release and viral shunt is relevant.

Minor comments

The search for "microdiverse population" in Google Scholar retrieves 26 results. What lies behind the term? Please search your text with the word “microdiverse”, if you agree with the proposed correction.

ANSWER: In the new version of the manuscript, to avoid confusion, we have replaced all “microdiversity” terms by “diversity”.

Check the provided sampling coordinates and their description again, or explain why the point 38.353972,0.426833 seems to be far from the point provided on the Figure 1. See also the compressed image in the attachment.

ANSWER:

We have corrected the coordinates to 38.35345133629738, -0.403225894080336. Thanks for catching the error

Take into account readability of the text on the figures and use standard spelling for the numbers.

Check if Figures 3, 4 and 5 were substituted with identical graphics by a mistake. It is especially hard for me to read the axis labels of Figures 4 and 5. Compare them to the labels on Figure 3.

Figure 5. “Number of virus”. Please check.

ANSWER:  We have modified the figures increasing the font size and readability of all text in general. In addition, we have corrected “number of virus” to “number of viruses”. Actually, we are not sure where was the mistake because in the system, we can indeed see different figures submitted.
